# Assessing Drug Administration Techniques in Zebrafish Models of Neurological Disease

**DOI:** 10.3390/ijms241914898

**Published:** 2023-10-04

**Authors:** Victoria Chaoul, Emanuel-Youssef Dib, Joe Bedran, Chakib Khoury, Omar Shmoury, Frédéric Harb, Jihane Soueid

**Affiliations:** 1Department of Biochemistry and Molecular Genetics, Faculty of Medicine, American University of Beirut, Beirut P.O. Box 11-0236, Lebanon; vc07@aub.edu.lb (V.C.); jb106@aub.edu.lb (J.B.); oas17@mail.aub.edu (O.S.); 2Department of Biomedical Sciences, Faculty of Medicine and Medical Sciences, University of Balamand, Kalhat P.O. Box 100, Lebanon; emanuelyoussef.dib@std.balamand.edu.lb (E.-Y.D.); chakib.khoury@std.balamand.edu.lb (C.K.)

**Keywords:** human neurological disease modeling, zebrafish, drug delivery systems, Alzheimer’s disease, Parkinson’s disease, traumatic brain injury, autism spectrum disorders

## Abstract

Neurological diseases, including neurodegenerative and neurodevelopmental disorders, affect nearly one in six of the world’s population. The burden of the resulting deaths and disability is set to rise during the next few decades as a consequence of an aging population. To address this, zebrafish have become increasingly prominent as a model for studying human neurological diseases and exploring potential therapies. Zebrafish offer numerous benefits, such as genetic homology and brain similarities, complementing traditional mammalian models and serving as a valuable tool for genetic screening and drug discovery. In this comprehensive review, we highlight various drug delivery techniques and systems employed for therapeutic interventions of neurological diseases in zebrafish, and evaluate their suitability. We also discuss the challenges encountered during this process and present potential advancements in innovative techniques.

## 1. Introduction

Zebrafish (*Danio rerio*) are rapidly emerging as a popular non-mammalian vertebrate model in scientific research. The exploitation of both the larval and adult stages has markedly expanded in neuroscience. Their high physiological and genetic homology to humans makes zebrafish a powerful model organism for the study of neurological diseases and the identification of potential therapeutic targets [1]. Zebrafish have gained popularity for large-scale screening due to their relatively small size and their capacity to generate large cohorts of genetically identical individuals quickly and efficiently [2,3]. This small size allows for the manipulation of hundreds of whole organisms on a single 96- or 384-well plate, which facilitates the efficient screening of a wide range of compounds at various developmental stages [4,5,6]. Female zebrafish can lay hundreds of eggs in a single clutch, which develop externally, making them easily accessible for experimental manipulation and providing researchers with a substantial sample size for experimental replication and highly significant statistical analysis [2,7]. Most organs, including the central nervous system (CNS), are fully functional within 48 h of post-fertilization [8]. Zebrafish embryos progress through their developmental stages swiftly [9,10], allowing researchers to observe and manipulate developmental and disease progression in a relatively short time frame [11,12]. The expanded lifespan of zebrafish (~4–5 years vs. 3 in mice) [2,13] allows, within a relatively short time frame, for the investigation of genetic and molecular mechanisms underlying aging and age-related disorders [14,15]. Zebrafish’s distinctive feature of transparency, particularly during the embryonic and larval stages of development, provides a unique opportunity to observe the development of various internal organ systems, tissues, and cellular processes in real-time without the need for invasive procedures that are commonly used in traditional mammalian research models [16,17]. This transparency permits high-resolution in vivo imaging, which provides valuable insights into the dynamic processes occurring within the developing zebrafish [18,19]. Techniques such as fluorescent labeling and calcium imaging have been used to visualize and track neural activity in zebrafish, aiding in understanding brain dynamics, disease progression, and drug response throughout embryonic development [20,21,22]. Zebrafish possess several homologous brain structures to those found in humans. This includes the telencephalon, diencephalon, mesencephalon, and rhombencephalon, to name a few [23,24]. Several cerebral nuclei in the zebrafish brain including striatum, basal ganglia, amygdala, and hippocampus display a high homology to mammals [23]. Key neurotransmitter systems found in the human brain, such as the glutamatergic, GABAergic, dopaminergic, serotonergic, and cholinergic systems, have been identified in zebrafish [25,26,27,28,29,30,31]. The D1 and D3 receptors in the dopaminergic (DA) system exhibit complete amino acid homology in the binding site between zebrafish and humans, whereas the D2 and D4 receptors share 85–95% homology [32].

Complete sequencing of the zebrafish genome by the UK Sanger Institute has revealed a high level of genome conservation between zebrafish and humans, with approximately 70% of their protein-coding genes in common [33,34]. This suggests that many fundamental biological processes and pathways are shared between the two species. Notably, 82% of disease-associated genes identified in humans have orthologs, or similar counterparts, in the zebrafish genome [33]. Among these are genes implicated in the pathology of many neurological diseases such as Alzheimer’s disease [35], Parkinson’s disease [36], Huntington disease [37], and amyotrophic lateral sclerosis [38,39]. Genome-editing technologies ranging from random mutagenesis screens [40] to targeted mutation using zinc-finger nucleases [41,42], TALENs [43,44], and more recently CRISPR/Cas9 [45] have resulted in numerous disease models. Generated zebrafish mutants exhibit phenotypes resembling human disease states [46,47,48,49] and provide high-throughput screening capabilities for evaluating drug delivery systems, assessing pharmacokinetics, and investigating potential drug toxicity [50,51,52]. By implementing anatomical techniques and behavioral tests (such as the novel tank diving test, the light/dark test, the social interaction test, the predator test, the avoidance test, and the T-maze test) [53,54,55,56,57,58], researchers have observed behavioral changes and modifications in neuronal connectivity in zebrafish models of many neurological diseases including Alzheimer’s disease [59,60], Parkinson’s disease [61,62,63], and autism spectrum disorders [41,64].

This review article, after highlighting the challenges encountered in zebrafish neurological disease research, will provide an overview of the main drug delivery techniques and systems in zebrafish, then focus on the use of these techniques in zebrafish neurological disease models of Alzheimer’s disease (AD), Parkinson’s disease (PD), traumatic brain injury (TBI), and autism spectrum disorders (ASD). Future advancements in drug delivery techniques will be discussed.

## 2. Challenges in Zebrafish Neurological Disease Research

The use of zebrafish as a model organism presents a significant number of advantages. There are, however, some limitations to its effectiveness in modeling human/mammalian neurological diseases.

Despite their homology, some brain regions such as the neocortex are not as developed in zebrafish as they are in humans [65]. This difference in complexity presents a disadvantage when analyzing complex behaviors such as changes in social and motor behavior, which reside in these exceedingly complex human brain structures. Moreover, some brain structures in zebrafish have not yet been accurately mapped and matched to their human counterparts. Some functions performed in the human brain by a singular region may often have several regions in a zebrafish brain, which function together to produce the same effect, and not all of these discrepancies have been characterized yet. This has left a gap in the understanding of how different brain regions and circuits interplay to produce a particular disease phenotype or physiologic state [66,67].

Another limitation of using zebrafish resides in the pharmacodynamic and pharmacokinetic differences between zebrafish and humans, but also in zebrafish themselves throughout their development. This poses significant hurdles when it comes to drug delivery/efficacy trials, as results may not be translatable to human subjects, or may be inaccurate and irreplicable when performed at different developmental stages [68].

Yet another drawback of this model organism resides in its limited genetic potential. Although humans and zebrafish share many clinically important and conserved genes, some zebrafish species possess gene alterations such as whole-gene duplications and deletions, which make assessing their functionality, or lack thereof, a challenge. Liu et al. showed that in zebrafish, the number of gene duplication sets goes up to almost 4000 [69]. Knocking out one copy of a gene but not the other may complicate and even skew the results obtained [66,70]. Zebrafish present a rarity of well-characterized inbred strains with some genetic and physiologic variability that remains however easily and readily accessible to researchers [71]. Despite efforts to mitigate this limitation, and the presence of certain genetically engineered strains, most breeding of zebrafish remains outbred, leading to an unclear breeding history and heterozygosity at several genetic loci, which may interfere with the studies conducted and the accurate interpretation of the results and their reproducibility [70,72].

An important challenge of using zebrafish to study the cellular and molecular pathophysiology underlying neurological diseases is the discrepancies that can arise when comparing results between zebrafish and rodents [73]. (The following reviews provide a comprehensive overview of the latest advances in zebrafish and rodent models of neurological disease [74,75,76,77,78,79,80,81,82]). These discrepancies can result from differences in the genetic background between both species. One such example is the study of the role of synuclein proteins in PD. In humans, misfolded α-synuclein proteins encoded by the SNCA gene form aggregates that eventually lead to the formation of Lewy Bodies, which are a hallmark of PD pathology [83]. Unlike mice, zebrafish lack α-synuclein, but instead they express β-, γ1-, and γ2-synuclein proteins that seem to play a role in dopamine homeostasis and in movement regulation [84,85,86]. These differences are confounded with another limitation that the use of zebrafish imposes, which is the lack of true “inbred strains”, whose mosaic genetic background often leads to a lack of reproducibility when jumping the species barrier towards rodents, for example. Although rodent populations undergo extensive screening and quality checks to validate the genetic “purity” of a line [87,88,89], the breeding practices of zebrafish populations are not optimized to achieve such a result, nor is screening routinely conducted in the same manner. Despite the clear advantage of generalizability that heterozygosity brings to the table, even that is more often than not checked in different genetically controlled rodent lines to maintain some level of oversight and control over the genetic conditions at play, an aspect which zebrafish biomedical research still lacks.

In addition, the pathologic mechanism involved is also highly dependent on the disease under study, and maybe more importantly, the particular chemical or genetic induction model of the disease that was chosen. A noteworthy example is the study of Alzheimer’s disease. A famous mouse line termed the Amyloid Precursor Protein 23 line (APP23) is a carrier of a mutated human version of the protein that is overexpressed sevenfold in these mice over its endogenous counterpart [90,91]. This particular line, as its name implies, owes its characteristic phenotype of AD to the amyloid hypothesis of Alzheimer’s pathophysiology. However, it has been determined that, pursuant to the accumulation of amyloid plaques, the insult continues as a result of a neuroinflammatory process of gliosis which appears to be restricted to the area of the plaques themselves [90,92,93], accompanied nonetheless by a more diffuse appearance of reactive hypertrophic astrocytes in brain regions that are plaque-laden [93,94]. Microglial activation in the brain has been largely characterized as a key pathological mediator of neuronal and synaptic death in the disease’s course, and many studies have experimentally demonstrated evidence supporting this hypothesis in humans and rodents alike [95,96].

When we look at similar genetic lines of Alzheimer’s in zebrafish, however, an interesting finding arises. Whereas neuroinflammation is mainly described as a pathological mechanism in rodents, some findings in zebrafish suggest that in certain circumstances, these inflammatory processes may play a contradicting and beneficial role to the one described in rodents, promoting neurogenesis and synaptic maintenance [60,97]. Oddly enough, some human studies have displayed similar contradicting results, highlighting the fact that disease pathophysiology may be much more complex than previously thought, and that results which may often appear to be contradicting a staple model organism may actually be shedding light on an important, complex, and overlooked aspect [98]. Thus, it becomes apparent that the challenges of using zebrafish in neurological disease research may eventually yield unanticipated results that more closely mimic humans. This will subsequently help researchers choose drug targets and mechanisms more wisely, so as to maximize benefit and reduce unanticipated confounding results or even harm to the organism.

Despite these limitations, which researchers should be aware of and which present challenges towards the scalability of results to the mammalian scale, the zebrafish remains a valuable model organism which complements other models in providing unique insights into physiological and pathological states.

## 3. Insights into Drug Administration in Zebrafish Model

When a drug is administered for therapeutic purposes, several factors must be taken into consideration pertaining to the drug’s biochemical properties; the targeted cells, tissues or organs; and the administration time window. The latter is especially important, as key cellular processes occur within a defined timeframe during development, and hence must be targeted accurately in order to achieve the desired results. A comparison of the main developmental processes that occur in the CNS between humans, rodents, and zebrafish is summarized in Table 1. In addition, the choice of the most suitable administration route is of utmost importance. Drugs can be administered via localized delivery or via systemic routes in an uncontrolled manner resulting in peaks of drug levels that are unsustainable over time. To enhance the therapeutic effectiveness, delivery systems have been developed in order to deliver drugs in a controlled manner over a time period, with a specific releasing rate and being directed towards specific targets. Here, we present different techniques commonly used in zebrafish models, ranging from direct administration techniques to targeted drug delivery systems (Figure 1).

### 3.1. Direct Administration Techniques

#### 3.1.1. Bath Immersion

Bath immersion is a widely spread method of drug delivery in small fish such as zebrafish models. Its popularity stems from its ease of administration and efficiency, as it simply consists of mixing the drug or therapeutic compound into the water medium in which the organisms are bathing. This creates a milieu in which the target compound is omnipresent and ready for uptake by the zebrafish, whether through skin absorption or ingestion [118].

Bath immersion, as a noninvasive drug delivery technique, provides an easy means to perform drug-mixing studies, as the array of therapeutic compounds may be placed altogether in the water where they can be readily uptaken [118]. However, this technique does not appear to be as simple as it sounds and presents some limitations [119,120]. Studies were undertaken to evaluate and more accurately assess the advantages and limitations of bath immersion. Dal et al. [119], in their recent work on zebrafish as a model to screen for anti-tuberculosis drugs, opted out of bath immersion, and used a direct injection into the circulatory system of the organisms. It was clarified by the researchers that they chose to disregard this popular method despite its aforementioned benefits due to the fact that it presents a plethora of disadvantages at the absorptive level. First, compounds that are hydrophobic react poorly in aqueous media and thus present a challenge to be absorbed via bath immersion. Secondly, it is impossible to predict or assess the precise concentrations of drugs delivered to the organism via bath immersion, due to a number of factors such as the physico-chemical properties of the drug, the behavior of the zebrafish, the duration of incubation, and the initial dose of immersion [119,121]. Guarin et al. [122] showed that bath immersion as a standalone method of drug delivery led to the lowest levels of absorption for all of the drugs. Moreover, four of the drugs had some delayed absorption, although this was restricted to the gastrointestinal system and the fact that it had poor distribution throughout the body [122]. An important element to note is the use of embryonic zebrafish eggs in the study, and thus the presence of the chorion, which has been reported in several papers as an absorptive barrier, which hinders drug delivery via bath immersion [123]. However, it has become apparent that bath immersion provides limited usefulness when it comes to many therapeutic compounds, notably lipophilic ones, which are required to pass the Blood–Brain Barrier (BBB) when treating models of developmental or central neurological diseases. Sometimes, the ill-advised use of soaking for compound delivery may yield inaccurate results. Paul et al. [124] evaluated the impact of moderate alcohol concentrations on zebrafish development using soaking at brief exposures (i.e., maximum of 2 h). According to Guarin et al., however, although the concentrations used were not high enough to significantly affect the developmental process, a longer exposure period for better absorption, or the use of microinjection for better BBB penetrance, may have produced different results [122].

#### 3.1.2. Zebrafish Embryonic Microinjection

Zebrafish embryonic microinjection, or ZEM, consists of the injection of a drug, genetic material, or other therapeutic compound into the yolk sac using a microinjection needle. The dish of embryonic eggs is held in place with a contralateral hand as the needle is lowered to pierce the chorion and go into the yolk sac of the zebrafish embryo. Care and consideration should be taken to perform this step in one swift and smooth motion so as to avoid crushing or tearing the yolk sac, the chorion, or even injecting air bubbles into the embryo, all of which may prove to be lethal [125].

ZEM has been the preferred and predominant method of drug administration in embryos. This preference is mainly due to the fact that microinjection provides the distinct advantage of delivering a direct bolus dose at therapeutic concentrations that is almost immediately 100% accessible within the embryonic cell. This avoids many of the struggles encountered with the absorption and lipophilicity of drug compounds via other methods such as bath immersion [122,126]. Moreover, ZEM is the most efficient method commonly used to create genetically engineered knock-in or knock-out zebrafish [127]. This method has also proven quite successful for beginning an early course of treatment for a model of a developmental disease in order to ensure the treatment of a single precursor cell. Early in development, zebrafish have been shown to display an incomplete metabolism. Thus, their ability to process prodrugs/drugs may vary significantly as they develop, leading to false positive/negative results in drug screenings, especially when conducting continuous and long-term treatments. However, the drug to be tested may be isolated, metabolized properly by an adult organism’s mature liver enzymes in vitro, and subsequently re-injected into the embryo in its active form directly at the site of action [68]. Giusti et al. [68] have already developed such protocols for rat liver microsomes that have proven to be useful and promising. Although subsequent delivery modality could also be performed via bath immersion [68], microinjection presents many of the aforementioned advantages that appear to make it a better modality of choice. Another useful attribute of local microinjection therapy resides in better toxicity assessments. When a compound is delivered via microinjection to the organism at a known concentration, a lethal concentration N (LCN) may be determined, with N being the percentage of organisms that died as a result of the treatment at that particular dose. Zhu et al. clearly demonstrated this in their work, while also highlighting the fact that other methods such as soaking fail to provide such data, as the amount of a drug taken up by the organism would be largely dependent on many factors which cannot all be accounted for, such as the duration of exposure, the starting concentration, the behavior of the organisms in the water, and the compound’s chemical and physical characteristics [122,128].

Although microinjection has proven extremely useful in many circumstances, it has many drawbacks to consider. One such issue is the limited time frame during which microinjections are the most effective in many cases, which is the one-cell embryonic stage [125]. This, however, no longer becomes an issue when the treatment consists of localized tissue-specific drugs for adult zebrafish in the setting of acquired injury such as TBI [129]. Moreover, microinjection is a procedurally tedious and complex technique, which must be perfected in order to be executed with no damage to the organism being injected. Many studies are currently underway to develop a fully automated and robotic microinjection procedure, in order to standardize, streamline, and simplify the process [130]. Guarin et al. showed that, although microinjection generally provides a better distribution of the injected materials across the entire body of zebrafish than soaking, this is largely dependent on an interplay between the physicochemical properties of the compound in question, as well as the site of microinjection [122]. For example, the yolk sac microinjection of a lipophilic compound would lead to a slower distribution (slow-release effect) into the circulation than a hydrophilic compound. This is because the lipid nature of the yolk sac would retain some of the compound until its plasma concentration decreases enough for more distribution to occur. Yolk sac microinjection in particular also led Zhu et al. to the unexpected finding that certain non-cardiotoxic drugs displayed a significant toxicity to treated zebrafish only upon this method of administration, highlighting yet another potential limitation of using microinjection with certain compounds [128].

#### 3.1.3. Oral Administration

Oral drug administration in zebrafish models is a common method used to study the pharmacokinetics and pharmacodynamics of various compounds [131,132,133]. Food-based drug delivery systems involve encapsulating drugs or therapeutic agents within feed pellets or microcapsules that are consumed voluntarily by adult zebrafish [134,135]. The screening of small molecules in adult zebrafish is challenging due to the inaccurate dosage when combining them with fish food or dissolving them in water [136,137,138]. To overcome the hurdles of drug testing in zebrafish, different methods of oral drug gavage have been developed with the maximum volume administered not exceeding 1% of the fish bodyweight [139,140]. Kulkarni et al. [131] were the first to conduct oral drug administration in adult zebrafish using a micropipette with a small tip that is gently pushed down the esophagus to about 1 cm from the mouth opening, while ensuring that no spillage occurred through the oral cavity or gill filaments. Successful drug delivery was confirmed by measuring the drug blood levels [131]. In order to avoid the regurgitation of the administered solution, Dayal et al. [141] slightly modified the oral administration technique implemented by Kulkarni et al. [131] by using a butterfly needle which expedites the drug passage into the esophagus. A study by Cocchiaro et al. introduced a needle-based micro gavage in zebrafish larvae using embryonic microinjection and stereomicroscopy equipment for the direct intestinal loading of substances [138,142,143]. The authors evaluated the efficacy and toxicity of this method and found that a large number of animals (seven to ten fish) can be effectively manipulated per minute, with high survival rates approaching 100% [142].

Although the safety of the gavage process is verified by using fluorescent microspheres, these delivery methods require skill and experience to be performed correctly without the risk of damaging the fish’s internal organs [131,142]. Passive drug delivery systems are ineffective and expensive for poor water-soluble substances [144]. An oral drug gavage regimen allows for precise dosing, which enhances the accuracy and reproducibility of experimental results [145]. This controlled drug delivery method reduces the hurdles of high variability in voluntary consumption by the fish that are encountered in other drug delivery systems. Moreover, oral gavage is a non-invasive method of drug administration that enables longitudinal studies of the effects of drugs or chemicals on various biological processes [143]. Recurrent anesthesia in zebrafish for long-term oral drug delivery studies presents several limitations and risks that researchers must carefully consider. Frequent anesthetic exposure can lead to cumulative stress [146], altered drug metabolism and absorption making it challenging to interpret drug effects accurately [147], and anesthetic tolerance, where repeated exposure to the same anesthetic agent can reduce its effectiveness over time. As a consequence, higher doses may be required to achieve the desired sedation level, potentially exacerbating the risk of toxicity, and adversely impacting the study’s outcomes [148,149,150]. A continuous monitoring of the zebrafish during experiments is essential to identify any signs of distress and ensure the safety and reliability of research outcomes.

### 3.2. Targeted Drug Delivery Systems

#### 3.2.1. Genetic Manipulation Techniques

Genetic manipulation techniques have proven to be very useful tools in creating zebrafish models for a number of neurological disorders [151,152,153]; however, there is relatively little exploration of their use as therapeutic agents in an attempt to reverse or reduce the disease phenotype in animal models. In the following, we will present two main gene editing methods in zebrafish, morpholinos and CRISPR/Cas9, and will briefly discuss their therapeutic potential in the context of zebrafish and disease research.

Morpholino oligonucleotides, with their ease of use and low cost, were, until recently, the most widely used antisense knockdown tools for zebrafish. Morpholinos are stable, nonionic, and water-soluble antisense oligonucleotides that bind complementary RNA target sequences via Watson–Crick base-pairing. They possess backbone linkages that are distinct from those of RNA and DNA, rendering morpholinos resistant to nuclease activity. The nonionic nature of the morpholino structure prevents the interaction of morpholinos with proteins, thus allowing for greater target specificity. Morpholino binding to the target RNA sequence blocks mRNA translation, pre-mRNA splicing, and miRNA maturation and activity. Importantly, because they are not recognized by proteins, morpholinos do not activate RNase H or the RNA-induced silencing complex (RISC); consequently, morpholinos do not lead to RNA cleavage and degradation [154,155]. Morpholinos can be used to alter pre-mRNA splicing patterns to yield different protein functional variants. This can be achieved by targeting morpholinos to splice-site sequences to either include an intron or excise an exon from the pre-mRNA, leading to the production of different protein variants [155]. This technique has been utilized in the treatment of Duchenne Muscular Dystrophy [156], attesting to the therapeutic potential that morpholinos possess. An additional utility of morpholinos exists in engineering zebrafish models of neurological disease such as models for ASD [157], Pitt–Hopkins syndrome [151], myasthenia gravis [152], and Batten’s disease [158]. Additionally, targeting viral RNA with morpholinos may preclude viral replication and serve as another therapeutic modality [155]. Finally, radio-labeled morpholinos can be used to target specific tissues for imaging studies, subsequently reducing radiation exposure to the organism [155]. Due to their water-soluble nature, morpholinos do not easily cross the plasma membrane; however, a variety of techniques can be utilized to overcome this hurdle. Scrape loading, electroporation, peptide conjugated-morpholinos, lipofectamine, Endo-Porter (an amphiphilic peptide) [155], and exosomes [159] are all modalities that increase the permeability of the cell membrane to morpholinos and are usually used in the context of cell cultures. The microinjection of morpholinos seems to be the most common delivery mechanism in zebrafish and allows for the direct delivery of oligonucleotides into the cytoplasm. In all cases, the delivery of morpholinos into the cytosol is often sufficient for RNA targeting because unmodified morpholinos can diffuse into the nucleus from the cytosol [155]. Morpholinos may be toxic to the organism above certain effective concentrations, further emphasizing the need to take caution in oligonucleotide administration [154].

Genome editing was revolutionized with the advent of the CRISPR/Cas9 system [160,161], which became the most widespread technique in most model systems. Clustered Regularly Interspaced Palindromic Repeats, or CRISPR, is a prokaryotic genetic editing system that was adapted for research purposes by coupling a single-guide RNA (sgRNA) to a CRISPR-associated protein (Cas9) to obtain what is the most accurate method of genome editing to date. The sgRNA is composed of a fusion of a CRISPR RNA (crRNA) that specifies the target sequences for the Cas9 protein, and a transactivating CRISPR RNA that serves the function of a binding scaffold for the Cas9 protein. Once the sgRNA binds the target sequence, the Cas protein recognizes a protospacer adjacent motif (PAM) and catalyzes the double-stranded cleavage of DNA, subsequently allowing for non-homologous end-joining or homology-directed repair to occur [162,163,164]. A variety of methods exist for delivering CRISPR/Cas9 into a cell. Plasmids expressing the Cas9 protein and the sgRNA can be electroporated into cells to confer the desired effects; however, this method results in the continuous expression of the complex and may increase off-target effects. In contrast, the microinjection of a modified CRISPR ribonucleoprotein complex encapsulated in lipid nanoparticles allows for more specific base-editing in zebrafish [165]. CRISPR/Cas9 has been utilized to generate zebrafish models of ASD [64], epilepsy, and other disorders of brain development [153]. Importantly, discrepancies can be observed between models generated via gene expression knockdown with morpholinos as opposed to knockout with CRISPR/Cas9, rendering the latter a more reliable genetic engineering tool in zebrafish [153,157]. A number of studies attest to the validity of using CRISPR/Cas9 in a therapeutic manner. Rees et al. utilized CRISPR/Cas9 homology-directed repair to rescue a zebrafish albino phenotype [166]. Moreover, Liu et al. used a different approach to rescuing a different zebrafish albino model by utilizing the CRISPR/Cas9 system to repair a premature stop codon [167]. Li et al. describes using a modified CRISPR system that induces precise germline mutations with high efficiency in zebrafish which provides an avenue for germline therapeutics to prevent disease manifestation in the progeny [168]. In an attempt to reverse the retinitis pigmentosa (RP) phenotype in an EYS (Eyes shut homolog)-associated RP zebrafish model, Schellens et al. [169] employed CRISPR/Cas9 technology to generate an excised form of the EYS gene. This strategy was finally not proven efficient as a potential treatment for RP due to inter-species differences between humans and zebrafish. Whilst transcript levels of the target sequence returned to normal, the expression of the implicated protein could not be detected because of the presence of translation regulatory elements in the deleted intronic regions [169]. This particular study indicates the need for caution when approaching therapeutic genetic manipulation. Other limitations reside in the fact that the Cas9 protein can only cleave the target if a specific PAM sequence is recognized, limiting the use of CRISPR/Cas9 to specific situations [162]. Additionally, CRISPR/Cas9 may be associated with off-target effects, incomplete editing, or inaccurate on-off target editing [170]. Finally, a major challenge in CRISPR therapy is the efficient delivery of the CRISPR/Cas9 complex in cells in vivo at levels sufficient for conferring the therapeutic effects.

#### 3.2.2. Nanoparticles

Nanotechnology-based drug delivery systems are relatively novel but promising techniques used in studying the pharmacological properties of drugs. Nanomaterials with sizes ranging between 1 and 100 nm are mostly used as carrier agents to encapsulate drugs or attach them to specific targets, allowing for their selective and controlled delivery. Several nanoparticles (NPs) were developed for drug delivery, as they can serve as nanocarriers, chaperones, or simply affect specific tissues with precise dosing and targeting [171]. Nanotechnology encompasses a wide array of techniques and substances, including metal and metal oxide NPs [172], nanocrystals [173], polymeric NPs and nanocapsules [174], liposomes [175], and nanogels [176], to name a few [171]. NPs can transport drugs of low solubility, perform a controlled drug release with a targeted delivery, manipulate local accumulation and biodistribution, and offer real-time, high-quality visualization opportunities [171]. NP-based drug delivery application in the medical field, called nanomedicines, has become an ideal technology for optimizing drug concentrations in specific tissues without affecting off-target organ systems [177]. Nanomedicines are developed to have a programmable stimulation-dependent drug release, further decreasing the risk of toxicity and adverse effects [178]. In neurodegenerative diseases, this translates as delivering drugs directly to the brain as well as enabling the CNS accumulation of desired drugs, which has been a challenge among researchers for decades now. The biggest hurdle in this process is the drug’s ability to cross the BBB, which allows only the entry of very small and lipid soluble molecules [179]. Nanotechnology-based drug delivery has become a focus area within the research community for developing therapeutic nanoparticles that overcome the BBB and transport neuro-specific drugs to targeted areas of the brain. In this context, zebrafish have emerged as an excellent model to assess NP efficiency and toxicity. To overcome the poor solubility of resveratrol (RES), by using a non-flavonoid polyphenol extracted from plants, which exhibits neuroprotective activities against Parkinson’s disease (PD), Xiong et al., (2020) developed a nanocrystal formulation (RES-NCs). This formulation had no significant toxic effects on zebrafish embryos and larvae, and led to an increase in RES oral absorption and a greater CNS bioavailability and brain uptake compared to RES alone [173]. Complexing nanoparticles with polymers permits precise target delivery and accumulation levels through complexing polymer chains with compounds expressing preferred chemical properties [180]. Rabanel et al. [174] investigated the use of PEGylated polyester NPs in order to enhance translocation across the BBB and to ameliorate neuronal uptake. They demonstrated that PEG chain density, length, and properties influence the endocytosis rate of nanoparticles through the BBB as well as their brain bioavailability [174]. Recently, cubosomes, which are non-lamellar lipid liquid crystalline nanoparticles, have gained interest as drug delivery vehicles, especially in cancer therapy [181]. Cubosomes are nanoparticles formed from a lipid cubic phase and outer corona polymer that confer high stability. They can be engineered in vitro to selective pore sizes and targeted membrane receptors [182]. The efficiency of cubosome-mediated drug delivery across the BBB was investigated in the adult zebrafish model. Azhari et al., (2021) showed that the use of cubosome nanocarriers stabilized with Tween 80 enhanced drug delivery and brain penetration [183].

The toxicity risk of a nanomaterial is the main challenge encountered when designing a nanoparticle-based drug delivery system. NP toxicity classification depends on several factors ranging from the NP bioproperties like size, surface charge, and chemical groups, as well as several other factors such as zebrafish developmental stage, treatment duration, and treatment dose. In fact, nanoparticles have been extensively investigated as a means to induce neurological toxicity in zebrafish models to efficiently manifest neurodegenerative diseases and thus permit the study of preventive and reversing therapies [184,185,186,187,188]. Nevertheless, in vivo NP toxicity mitigates their biomedical applications as therapeutics [189]. For instance, silver NPs (AgNPs) have been found to induce oxidative stress in adult zebrafish, in addition to the generation of a genotoxic effect, which is the presence of micronuclei and nuclear abnormalities [190]. Zebrafish embryos exposed to gold NPs (AuNPs) showed perturbed inflammatory and immune responses [191]. One way to overcome this challenge is to use NPs as adjuncts to different systems. By combining nanotechnology, microinjections, and genetic manipulation, Patton et al. [175] showed an increased uptake, survival, and translation of lipid-nanoparticle-packaged, green fluorescence protein-tagged mRNA in an embryonic zebrafish model. This study showed little to no evidence of toxicity [175]. Kalaiarasi et al. [176] made use of nanogels’ properties, which can encapsulate great drug loads, have high BBB permeability, and allow for stimulus-dependent release. They showed a thermo-responsive and sustained release of donepezil when coated with poly N-isopropyl acryl-amide nanogel (DCN), which led to significantly less toxicity in zebrafish neurological models [176]. Overall, nanotechnology boasts multiple advantages and significant potential as a drug delivery system for the treatment of diseases once case-by-case toxicity characteristics are established [192].

#### 3.2.3. Hydrogels

Hydrogels have become increasingly popular as a promising class of materials in many biomedical fields, and their development and use have been the subject of several studies. Hydrogels are the favored wound dressings in clinical settings because of their exceptional mechanical and biochemical qualities. They offer a customizable approach to wound healing because they contain bioactive molecules and can be easily tailored to specific wound requirements. In addition, hydrogels act as cleansing systems by absorbing exudate and removing pro-inflammatory molecules from the wound surface [193]. Novel hydrogel scaffolds with an interconnected porous microstructure, mechanical strength, good hydrophilicity, and suitable biocompatibility behavior have been developed for tissue regeneration applications [194]. The use of injectable hydrogels was investigated in the central nervous system (CNS) in order to create an environment conducive to axon growth [195]. Additionally, hydrogels have been successfully used for neuronal repair, motion sensing, spinal cord injury repair, and sustained drug release, demonstrating their versatility in addressing various biomedical challenges [176,196,197].

Hydrogels offer many advantages for the administration of drugs in zebrafish. First, they have exceptional mechanical and biochemical properties that make them useful as dressings in clinical practice. This benefit was demonstrated by Corrales-Orovio et al., who focused on the potential of photosynthetic hydrogels as dressings to improve wound healing [193,198]. Hydrogels can be easily tailored to specific requirements, allowing for customization in drug delivery applications. They can deliver biomolecules to local tissues and promote wound healing by containing compounds like growth factors, antibiotics, and therapeutic drugs. The release of molecules can be controlled by customizing the hydrogels’ mechanical properties and pore size [199]. In addition, hydrogels act as cleansing systems by absorbing excess exudate, metabolic waste, and pro-inflammatory molecules on the wound surface, facilitating autolytic debridement [200].

Utilizing hydrogels as a drug delivery system in zebrafish has both limitations and drawbacks, despite the benefits. A limitation is the stability of physical hydrogels under physiological conditions, as they may be slow to respond to environmental stimuli [195]. This limitation was addressed by Hasanzadeh et al. [195], who focused on injectable hydrogels in the tissue engineering of the central nervous system (CNS). While injectable chemical hydrogels offer excellent mechanical properties and stability, they can have adverse effects in vivo due to chemical interactions [195]. Although several injectable biomaterials have shown promise in CNS regeneration in vivo, more research is needed to treat CNS damage. Synthetic biomaterials have lower biological activity and biocompatibility, despite their good mechanical properties and stability. Thus, developing hydrogel-based drug delivery systems for zebrafish should consider these limitations and focus on improving stability, biocompatibility, and functionality in specific applications like CNS repair.

A summary of the different modes of drug administration described in this paper is presented in Table 2.

## 4. Drug Delivery Techniques Used in Zebrafish Neurological Disease Models

### 4.1. Alzheimer’s Disease

With the developments in science and medicine over the past couple of decades, the world has witnessed a shift in the prevalence and burden of illnesses, moving us from preventable and infectious diseases towards a preponderance of chronic disorders known as “diseases of old age”. One such disease, which has been projected to affect more than 13 million in the United States by 2050, is Alzheimer’s disease (AD). It is a neurodegenerative disorder characterized as the most common cause of dementia in the elderly [201,202]. It has several types, with the early-onset variant presenting usually as a result of a genetic mutation in Alzheimer’s disease susceptibility genes, such as amyloid precursor protein (APP) or presenilin 1/2 (PSEN1/2). Late-onset AD has been shown to be a result of both genetic and environmental risk factors interwoven together to lead to the disease phenotype [203,204]. The latter has also been referred to as ‘sporadic AD’. The pathology of Alzheimer’s disease has two main pillars: neurofibrillary tangles (NFTs) and amyloid plaques (APs). The former is the result of defects to the tau protein, a microtubule-associated protein (MAP). Several etiologies, from conformational changes of the protein due to a mutation, to neuronal hyperexcitability and calcium signaling changes, all seem to lead to a common result, the hyperphosphorylation of the tau protein [204,205,206,207]. This in turn not only destabilizes the microtubules of neurons, but also leads to the aggregation of these MAPs into insoluble intracellular inclusion bodies that are known as NFTs, and ultimately to neuronal death [208]. Aps, on the other hand, are extracellular aggregates of another protein, the amyloid beta protein (Aβ). Upon cleavage of the amyloid precursor protein (APP) by enzymes such as ⍺ and γ-secretins, a decreased or absent clearance of the Aβ product leads to their aggregation and the formation of APs, sometimes referred to as senile plaques [204,205,209,210]. Some have hypothesized that the formation of senile plaques may be the initiating factor which precipitates NFT formation, and the remainder of the pathophysiological findings seen in AD patients’ brains [201,209]. These pathological mechanisms culminate in a number of symptomatic findings, such as dementia, sensory and motor deficits, cognitive changes, and memory impairment, among others [211,212].

Zebrafish have been used extensively in the creation of animal AD models to better understand the pathology of the disease. Alzheimer’s in zebrafish has been induced via both neuropharmacological [213,214,215,216] and genetic means [91,217,218,219]. Such models have been invaluable in the conduct of research geared towards finding targets for treatment. In their recent work, Nery et al. used bath immersion to treat AD models of zebrafish with lithium and were successful in reversing the animal’s tau hyperphosphorylation and cognitive deficits [216]. In 2019, Koehler et al. [220] used 4-benzyl-2-methyl-1, 2, 4-thiadiazolidine-3, 5-dione (TDZD-8), an inhibitor of the AD drug target Glycogen synthase kinase 3 β (GSK3β) to reverse cognitive impairment in a zebrafish model of the disease. Both Okadaic Acid and TDZD-8 were dissolved in the nutritive aqueous medium and delivered via soaking, and the treatment duration was 9 days [220]. That same year, Javed et al. [221] worked with casein-coated gold nanoparticles (betaCas AuNPs), and found that treatment with these nanoparticles recovered the mobility and cognitive functions of Aβ-induced adult zebrafish models of AD. A successful and efficient BBB crossing and drug delivery was confirmed via TEM imaging, ROS assays, and plasma-mass spectrometry, among others [221]. Yang et al. [222] synthesized a class of compounds known as 3-arylcoumarins and were able to show that these chemicals displayed high anti-acetylcholinesterase and anti-monoamine oxidase activity levels, shedding light on a new class of potential candidates for AD treatment. Their work also attested to the efficacy of using zebrafish as a biological tool for high-throughput pharmacological screening assays [222].

### 4.2. Parkinson’s Disease

Parkinson’s disease (PD) is the second most common neurodegenerative disease in the United States [223]. It is related to a loss of dopaminergic (DA) neurons of the substantia nigra in the midbrain and the development of Lewy Bodies. Its most prominent motor symptoms include resting tremors, bradykinesia, rigidity, a shuffling gait, and postural instability. Non-motor symptoms include cognitive impairment (dementia), depression, anxiety, autonomic dysfunction, and sensory/sleep disturbances [224]. The etiology of PD cases is the result of the combination of environmental and genetic factors. The risk of PD may be increased by head injury and toxicant exposure. Both genetic and lifestyle factors may modify how exposure to the environment affects an individual [223]. Genes thought to be causing PD include ⍺-Synuclein (*SNCA*), leucine-rich repeat kinase 2 (*LRRK2*), Parkin (*PARK2*), and phosphatase and tensin homolog (*PTEN*)-induced putative kinase 1 (*PINK1*). The mainstream PD therapies only provide symptomatic relief rather than slowing or stopping the neurodegenerative process of PD, making PD an incurable condition at the moment. This demonstrates the critical need for a thorough understanding of PD pathogenesis and the creation of new therapeutic approaches that can halt the progression of PD [188]. Parkinson’s disease is the most researched movement disorder using zebrafish [65]. Zebrafish have a dopaminergic neuronal system that has been extensively studied and contains orthologs for about 82% of all human disease genes [33]. When compared to human PD cases, the PINK1 deficient zebrafish model displays typical parkinsonian pathologies such as the loss of DA neurons and complex 1 inhibition [225]. In this zebrafish model of PD, it was shown that the inhibition of mitochondrial calcium uniporters (MCU), located in the inner mitochondrial membrane, using morpholinos and ruthenium red, rescues the dopaminergic neurons [226].

The BBB proves to be a significant obstacle in finding a treatment for PD, as achieving the required CNS accumulation of therapeutic agents requires bypassing this membrane without causing systemic or local toxicities. The use of nanotechnology has challenged this barrier with increasing progress. An early study by Nellore et al. found that platinum nanoparticles coated with leaf extracts from Bacopa monnieri (BmE-PtNPs) had a neuroprotective effect on MPTP-induced Parkinsonism in zebrafish animal models [227]. The crystallization of drugs has also been shown as effective by Xiaong et al. This study developed Puerarin nanocrystals (PU-NCs) that demonstrated a much higher permeability and cellular uptake than PU by itself, as well as exerting neuroprotective effects on 1-methyl-4-phenylpyridinium ion (MPP+)-induced Parkinsonism in the zebrafish model, all the while showing no toxicity [228]. Also using Puerarin, Chen et al. [229] encapsulated PU in poly(lactic-co-glycolic) acid (PLGA) nanoparticles, which resulted in an extended in vivo half-life, enhanced bioavailability, and greater CNS accumulation than PU alone. Their carrier system exhibited improved cellular permeability and uptake, as well as better neuroprotective effects [229]. Similarly, Wang et al. [230] developed poly(ethylene glycol)-poly(trimethylene carbonate) (PEG-PTMC) nanoparticles to encapsulate Ginkgolide B (GB) doses, showing a greater oral bioavailability and CNS accumulation than if given alone. These particles readily crossed the BBB, and showed high stability and negligible toxicity in zebrafish models [230]. Zhao et al. [231] highlighted the increased oral bioavailability of Ginkgolide B (GB) across the BBB when packaged in poly(ethylene glycol)-co-poly(ε-caprolactone) (PEG-PCL) nanocarriers. These NPs enabled a sustained release of GB and its efficient accumulation in the CNS [231]. Chen et al. [232] used Pluronic P85/F56 micelles as endocytic carriers to both cross the BBB and bypass the effect of multidrug resistance protein 2 (MRP-2). Their system successfully enhanced Baicalein cellular uptake and permeability, showing significant neuroprotective effects in the neurotoxin 1-methyl-4-phenyl-1,2,3,6-tetrahydropyridine (MPTP)-induced zebrafish PD model [232]. Due to its specificity and precision, microinjections have been studied as efficient drug delivery systems for studying PD in zebrafish models. Li et al. used intraperitoneal injections to efficiently deliver Acteoside in the neurotoxin 6-hydroxydopamine (6-OHDA)-induced zebrafish PD model, exhibiting decreased oxidative stress via activation of the Nrf2-ARE signaling pathway [233]. N-acetylcysteine (NAC) immersion has been used successfully by Benvenutti et al. to prevent locomotor and behavioral deficits in the 6-OHDA-induced zebrafish model [234]. Cronin and Grealy used rasagiline and minocycline immersion of 6-OHDA-induced zebrafish models to achieve neuroprotective results. Rasagiline and minocycline reversed neuronal apoptosis and locomotor deficits [235].

Nanoparticle research has demonstrated neuroprotective effects and enhanced drug delivery to the CNS in zebrafish models. Additionally, microinjections and immersion methods have proven effective in delivering therapeutic agents and preventing locomotor deficits in PD zebrafish models. These advancements in zebrafish research contribute to a better understanding of PD pathogenesis and the development of potential therapies.

### 4.3. Traumatic Brain Injury (TBI) and Stroke

Traumatic brain injury (TBI) is a sudden and brutal insult [236] to the brain caused by an external mechanical force that results in temporary or permanent impairment [237]. TBI is a debilitating global health problem with over 69 million people suffering from it each year, with an age range including the young population as well as the elder one [238]. Following the primary injury, a cascade of irreversible events occurs, leading to further damage to the brain tissue, including swelling, damage to the blood vessels resulting in a decrease in cerebral blood flow, hemorrhage, and long-term cognitive deficits [237]. While TBI and stroke are two distinct neurological conditions, studies have revealed an increased risk of stroke during the 3-month to 5-year follow-up period in individuals who have sustained a TBI [239,240,241]. Hence, prevention of these sequelae is the utmost target in the therapeutic aim of TBI.

Various drug delivery systems have been used to deliver therapeutic agents to the brain in zebrafish models of traumatic brain injury and stroke. To overcome the hurdle of crossing the BBB, nanotechnology-based drug delivery systems were developed to transport neuro-specific drugs to the targeted injured area of the brain [179]. Neuroprotective effects of hesperetin nano-formulations have been studied in a traumatic brain injury model of zebrafish. Paramita et al. [242] reported that the oral administration of nano-formulated hesperetin (nHST) in an adult zebrafish model of traumatic brain injury shows an antioxidant effect and an increase in catalase levels in brain tissues. In addition, treatment with nHST reduces the infiltration of inflammatory cells [242]. Due to its accuracy and efficiency, injection remains the most commonly used method of drug delivery in zebrafish models of TBI and stroke. A study conducted by Liu et al. revealed that retro-orbital venous injection of human umbilical cord perivascular cells in an adult zebrafish model of TBI presented reduced reactive gliosis and apoptosis, and an improvement in locomotor activity and anxiety [243]. An intraperitoneal injection of saffron in a stab wound adult zebrafish model of TBI reversed the cognitive and behavioral deficits observed in lesioned zebrafish [129]. Bath immersion has been used successfully in some studies to deliver drugs to zebrafish models of TBI and stroke. The bath application of glutamic acid induced a neurological injury that mimicked the traumatic mechanical insult of TBI in larvae zebrafish. The immersion of glutamate inhibitors delayed excitotoxicity, increased survival rate, rescued locomotor impairment, and reduced apoptosis [244]. Following hypoxia induced by the oxygen absorber, larvae zebrafish were immersed in an E3 medium containing the glutamate antagonist MK-801 and the free radical scavenger edaravone [245]. The Ponatinib (PON)-induced zebrafish ischemic stroke larvae model was immersed in a Guhong injection (GHI) composed of safflower extract and aceglutamide. The GHI showed an anti-thrombotic effect, ameliorated behavioral defects, and regulated the inflammatory pathway [246]. Crilly et al. [5] conducted an in silico screening of 150 drug compounds with structural similarities in the larval zebrafish model of spontaneous intracerebral hemorrhage. Using PREDICT and MBiRW algorithm datasets, six drugs were identified to reduce brain cell death and were predicted to interact with cerebrovascular diseases. Out of these six drugs, the bath immersion of larvae zebrafish in two angiotensin-converting enzyme inhibitors (ACE-Is), ramipril and quinapril, reduced cerebral edema and brain cell death [5]. While bath immersion is a non-invasive method of drug delivery, it has some limitations including drug stability and dosage. Drug delivery systems in zebrafish models of TBI and stroke face a number of challenges that should be considered carefully in the context of the research question that is being addressed and the properties of the drug being delivered.

### 4.4. Autism Spectrum Disorders

ASDs are a heterogenous and highly hereditary range of neurodevelopmental disorders characterized by atypical and repetitive behaviors, an impairment in social communication and interaction, and sensory anomalies, and they may also be associated with intellectual disability as well as developmental variations in neuroanatomy [247,248]. Environmental factors such as parental age, obesity, and diabetes are associated with the development of ASD [249]. Globally, ASDs affect 28.3 million people, with higher incidence rates in males as opposed to females [250]. Hundreds of genes have been implicated in the development of ASD [248]. Recently, Meshalkina et al. found that 62% of the 858 human ASD risk genes listed in the Simons Foundation Autism Research Initiative (SFARI) database have zebrafish orthologs [251]. Zebrafish models of ASD are typically generated via the knockout or knockdown of a single gene [157]. Fragile X syndrome, an ASD-associated syndrome caused by mutations in the FMR1 gene, is a leading genetic cause of autism in humans. ENU (N-ethyl-N-nitrosourea) mutagenesis has been utilized to generate knockdown models of FMR1 [252], resulting in zebrafish with ASD-like behavior [157]. Other monogenic zebrafish models of ASD involve the knockout of *arid1b* [253], *chd8* [254], *shank3b* [64], *mecp2* [255], and *setd5* [256], among others. Importantly, the monogenic nature of zebrafish models of ASD does not fully recapitulate the environmental influences or the typically polygenic nature of ASD in humans; however, it can provide a simplistic and targeted approach to studying the effects of different therapeutic modalities in reducing the ASD phenotype under specific conditions. A variety of social behavioral tests can be used to assess the behavior of ASD models of zebrafish, such as the shoaling test, the social preference test, and the kin recognition test [64,157,248]. The shoaling test is used to assess shoaling behavior, which is the formation of clustered groups in which fish swim together. Shoaling impairment indicates an impaired social interaction/cohesion in zebrafish [257,258]. Zebrafish models of ASD also exhibit impaired social behavior in the social preference test, which assesses the animal’s preference for a social stimulus over a non-social stimulus [259], and in the kin recognition test, in which zebrafish fail to recognize a kin group composed of conspecific fish of the same color [258].

Few attempts at rescuing autistic-like phenotypes were performed using compounds administered through simple water immersion. Gabellini et al. showed that bath immersion with risperidone rescued social interest behaviors in *setd5* knockout zebrafish [256]. Rahmati-Holasoo et al. found that bath immersion with oxytocin provided neuroprotective effects and improved social behaviors in a valproic acid-induced ASD model of zebrafish larvae [260]. Zhang et al. found that a minocycline bath immersion for 2 weeks reduces autism-like behavior in *nde1* knockout zebrafish [258]. CRISPR gene manipulation offers a wide spectrum of therapeutic approaches to rescue monogenic ASD phenotypes. For instance, CRISPR/Cas9 knockout of *fabp2* ameliorates social behavior in a zebrafish ASD model of maternal immune activation. Alternatively, Lie et al. utilized a modified CRISPR/Cas9 system to restore Fmr1 expression in fragile X syndrome-induced pluripotent stem cells (iPSCs) via epigenetic modification, subsequently restoring the neurophysiological properties and wild-type phenotype of derived neurons even after engraftment into mouse brains [261]. However, the efficacy of CRISPR epigenetic modification is yet to be demonstrated in zebrafish in vivo.

### 4.5. Amyotrophic Lateral Sclerosis

Amyotrophic Lateral Sclerosis (ALS), also known as Lou Gehrig’s disease or motor neuron disease (MND), is the most common disease of motor neuron degeneration with an incidence of 0.3–3.3/100,000 people affected every year [262], and a familial inheritance component of about 10% [263]. Whilst the hereditary component makes up about only 10% of ALS cases, there are yet to be any definitively proven risk factors for ALS [263]. ALS pathology is characterized by the degeneration of upper and lower motor neurons, the frontotemporal lobes, and neurons in other regions of the brain [263]. ALS typically presents in late middle life with progressive muscle weakness and atrophy, affecting the performance of even daily activities like walking or writing, and is characterized by clumsiness, slurred speech, trouble swallowing, and twitching, among other symptoms [264]. Respiratory inadequacy limits survival to 2–4 years after first presentation [264], ultimately leading to respiratory failure and death [263].

Whilst the foundational pathological mechanisms governing ALS neurodegeneration are not fully understood, a multiplicity of genes with different functions have been implicated [265]. Almost 97% of ALS cases have a defective DNA-binding protein 43 (TDP-43) gene, leading to the formation of cytoplasmic aggregate and the loss of function of the normal TDP-43. Another commonly defective gene in ALS is C9orf72, in which the loss of its proteins causes motor neuron death. The gene coding for super-oxide dismutase (SOD1) forms intracellular aggregates that halt protein degradation when mutated [266,267]. Fused in sarcoma (FUS) gene mutations are debated as to whether loss- or gain-of-function mechanisms lead to the deleterious effects seen in this subset of ALS patients. The accumulation of these protein complexes as well as the loss of function might mediate the propagation of cell death in ALS, presenting as motor neuron disease [266].

Zebrafish have high conservatism with humans in terms of genes, sharing orthologs with more than 70% of all human genes [33], including genes implicated in the pathology of ALS (SOD1, TARDBP, C9orf72, and FUS) [268], with these genes having been utilized to generate zebrafish ALS models. Zebrafish ALS models have been generated via the exposure of zebrafish larvae to cyanotoxins [269], through random mutagenesis via N-ethyl-N-nitrosourea to generate a SOD1 mutant model [38], or through genetic manipulation techniques involving transient gene overexpression via the mRNA microinjection of an incomplete C9orf72 construct [270], overexpression of SOD1 [271], morpholino knockdown of C9orf72 [270] or progranulin [272], and CRISPR-Cas9 [273,274]. Zebrafish ALS models also exhibit pathological and behavioral phenotypes comparable to those seen in human ALS patients. For instance, ALS models of zebrafish generated via SOD1 overexpression exhibit pathological changes in the neuromuscular junction, muscle atrophy, the loss of motor neurons, paralysis, and premature death [271].

The etiology of ALS has been shown to be incredibly complex, leading to a scarcity of adequate treatment options, with the only widely accepted treatment method for ALS being Riluzole therapy, with only mild efficacy [275]. Additionally, anti-inflammatory and anti-oxidative compounds have failed to show an improvement in disease progression in clinical trials, and almost all subsequent therapeutic techniques have failed to produce significant positive results [276]. However, experiments on zebrafish models of ALS provide a promising avenue for potential therapeutics. Goldshtein et al. have demonstrated that a ciprofloxacin and celecoxib bathwater immersion of SOD1 and TARDBP mutant zebrafish leads to a dose-dependent improvement of locomotor activity in both models, and an ameliorated motor neuron axonopathy in the SOD1 model [277]. Patten et al. showed that a pimozide (a narcoleptic drug) bath immersion improved synaptic transmission in the neuromuscular junction of TDP43 mutant zebrafish [278]. Following that, Bose et al. have shown that bath immersion with a novel small molecule TRVA242 pimozide derivative improves the ALS phenotype in loss of function tardbp, gain of function TDP43, and SOD1 mutant zebrafish embryos [274]. Chaytow et al. demonstrated that a bath immersion in Terazosin or a microinjection of phosphoglycerate kinase 1 mRNA in C9orf72 and TDP-43 mutant zebrafish ALS models ameliorated motor behavior and rescued the motor axon phenotypes [279]. Finally, Lattante et al. demonstrated that the injection of rapamycin into sqstm1 knockdown zebrafish ALS models improved the ALS motor phenotype [280]. Whether genetic manipulation techniques can be used to ameliorate the ALS phenotype in ALS models of zebrafish remains to be seen.

## 5. Concluding Remarks and Perspectives

The exploration of various drug delivery systems in zebrafish models for neurological diseases holds significant promise in advancing our understanding and treatment of these complex conditions. Zebrafish show genetic and structural brain homologies to humans, which offer a unique platform for studying drug delivery mechanisms and evaluating the efficacy and safety of novel therapeutic approaches. One of the key benefits of using zebrafish is the ability to perform high-throughput drug screening, allowing researchers to test a wide range of potential treatments rapidly. Furthermore, zebrafish provide a valuable in vivo system for studying the biodistribution and pharmacokinetics of drug candidates. However, challenges persist in implementing drug delivery systems in zebrafish models for neurological diseases. The anatomical and physiological differences between zebrafish and humans necessitate a cautious interpretation and extrapolation of the results. Scaling up findings from zebrafish to higher mammals, including humans, requires careful consideration of species-specific variations. Despite these challenges, the use of zebrafish models in studying drug delivery systems for neurological diseases presents an exciting opportunity to develop more targeted and efficient therapeutic drug delivery methods.

In addition to the recent focus on nanotechnology-based techniques used to efficiently cross the BBB and directly target specific regions and/or cell populations in the brain, other approaches are being investigated. Using the transient expression of defined groups of transcription factors, somatic cells (mostly fibroblast cells) can be de-differentiated to produce induced pluripotent stem cells (iPSCs), which, in-turn, can be programmed into any other cell type [281]. In the case of neurodegeneration, there is great interest in attempting to generate tissue replacement by producing specific iPSC-derived neuronal engraftment. iPSC-derived neuronal therapy has been successfully used in non-human primates [282] and rodents [261,283,284]. Human iPSC-derived neural precursors have been shown to survive and differentiate in zebrafish embryos for more than 2 weeks [285]. However, more research is required to elucidate the efficiency of iPSCs as a drug delivery or therapeutic modality in zebrafish models of neurological disease.

As research on drug delivery systems advances and our understanding of zebrafish physiology improves, this model organism is likely to continue playing a pivotal role in shaping the future of neurological disease treatment strategies.

## Figures and Tables

**Figure 1 ijms-24-14898-f001:**
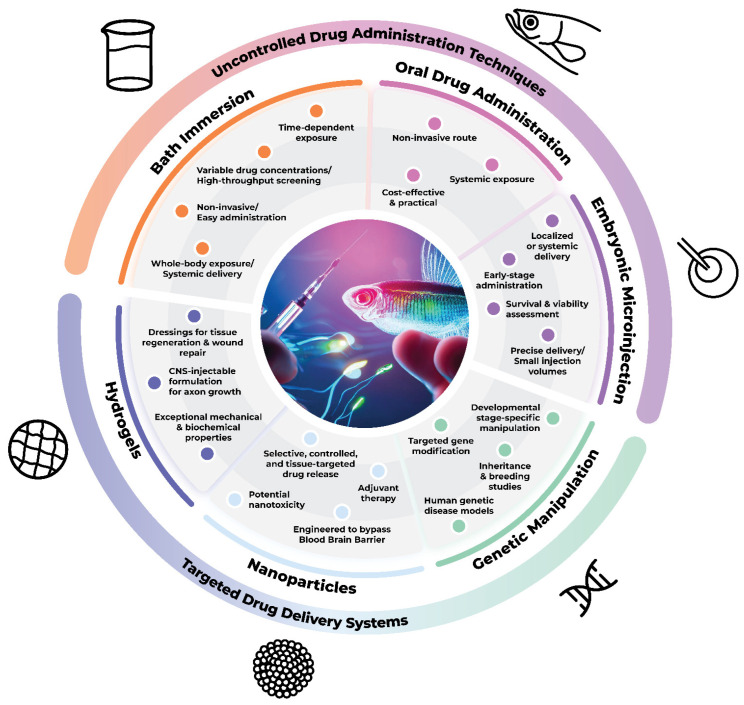
Techniques employed for drug administration in zebrafish and their distinctions concerning the mode of administration.

**Table 1 ijms-24-14898-t001:** Comparison of key developmental processes in the CNS in humans, rodents, and zebrafish.

Developmental Processes	Zebrafish	Rodents	Human	References
Establishment and maturation of BBB	3 dpf—10 dpf	Pnd 1–3	23–32 g.w.	[99,100,101,102]
Neurogenesis	10 hpf—adult	E10.5–E19.5	6 g.w.–2.5 y.o.	[103,104,105,106]
Peak in synaptic density/pruning	8 dpf (in optic tectum)	Pnd 20–35	2–3 y.o.	[107,108,109,110]
Astrocyte formation (observation of mature astrocytes)	10 dpf in brain parenchyma. 2–4 dpf in spinal cord	E60–E95	12–40 g.w.	[101,102,111,112,113]
Oligodendrocyte maturation	60 hpf	Pnd 0—adult (peak at pnd 14)	15–28 g.w.	[114,115,116]
Myelin development	60 hpf—adult	Pnd 7–90	28—adult	[116,117]

dpf: days post-fecundation; hpf: hours post-fecundation; pnd: post-natal days; E: embryonic stage; g.w.: gestational weeks.

**Table 2 ijms-24-14898-t002:** Different modes of drug administration in zebrafish models, with a description of the protocol used, the advantages, and the challenges encountered for each.

Mode of Drug Administration	Protocol	Advantages	Challenges
Direct administration techniques			
Bath immersion	Drug/therapeutic compound is directly dissolved into nutritive water media of the zebrafish.	- Easy to administer. - Allows investigators to conduct drug-mixing studies easily. - Useful when administering hydrophilic drugs targeting the GI tract.	- Exact concentration or time of drug administration cannot be determined. - Not cost-effective: higher concentrations of drugs needed to account for factors like dispersion and absorption efficiency. - Poor BBB penetrance.
Microinjections	- Embryo dish is held steady with contralateral hand. - Needle is inserted through the chorion into the yolk sac.	- 100% instantaneous bioavailability at injection site. - Allows conduct of toxicity assessments for compounds. - Better widespread distribution across the organism’s body as compared to other modalities.	- May damage organism if improperly carried out. - Tedious and technical procedurally. - Distribution efficacy may vary based on the nature of the compound and the chosen injection site. - Some cytotoxicity when used for certain compounds.
Indirect administration techniques			
Oral administration	- Micropipette method. - Needle-based method.	- Reduces variability in voluntary consumption by the fish. - Enables accurate drug delivery to experimental animals resulting in more accurate drug evaluations. - Improves bioavailability of poorly soluble drugs. - No trauma introduced by invasive injections, which allows for long-term daily treatments (longitudinal study).	- Risk of internal organs damage if not performed correctly. - Anesthetic toxicity and drug interaction.
Genetic manipulation	- Antisense Morpholinos. - CRISPR/Cas9 technology.	- Target-specific. - Allows genetic screening. - Allows controlled effect duration. - Can target virtually any sequence.	- Off-target effects. - Incomplete editing. - Limited delivery. - Can generate inflammatory responses.
Novel drug delivery systems			
Nanoparticles	- NP-chaperone. - Nanocarrier. - Pure NP.	- Target-specific (except pure NP). - Sustained release (nanocarrier). - Crosses BBB. - Visualization properties of carriers/chaperones. - Real-time measurement of accumulation and distribution.	- Toxicity (systemic and local) and teratogenicity.
Hydrogels	- Wound dressing. - Injectable.	- Mechanical and chemical stability with tunable degradation rate. - Nontoxic and with high drug-loading capacity and BBB permeability. - Imitates ECM features. - Highly hydrophilic and porous. - Sustained release of bioactive molecules topically. - Can be easily modified to meet requirements of diverse wounds. - Absorbs wound exudate. - Can be injected into specific areas with minimal invasiveness. - Resembles many aspects of the CNS, promoting its regeneration.	- Difficult to compare the oxygenation capacity of hydrogels with other types of wound dressing. - Act differently in vivo and in vitro. - Need to remove tissue from the damaged site to allow insertion of the hydrogel. - Hydrogels derived from pure natural polymers often have low mechanical properties and risk for immunogenicity.

## Data Availability

Not applicable.

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
