# Peer review of "Assessing Drug Administration Techniques in Zebrafish Models of Neurological Disease"

_ijms, 2023, doi:10.3390/ijms241914898_

Round 1
Reviewer 1 Report
This review provides a balanced overview of the advantages and limitations of using zebrafish models of human disease followed by an in-depth review of drug delivery methods currently used in zebrafish larvae and adults, which includes lots of specific examples and a summary table. The review then focuses on drug administration in specific neurological diseases (AD, PD, TBI and stroke and ASD) including novel therapeutic strategies such as nanoparticle delivery. The review addresses a clear gap in the literature where drug delivery techniques in zebrafish have not been reviewed before, so I think it contributes an interesting insight to the field and will be of interest to many researchers in this area. The use of zebrafish models in drug screening and the use of zebrafish to model neurological diseases is increasing, so I think this review is timely for those looking to learn more in this area.
Overall, this review is well written, balanced and comprehensive. However, there are a few minor points that need to be addressed before publication, which are listed below:
1.) There are multiple paragraphs which are close to a page long. I think it would improve the readability if these were broken down into shorter paragraphs.
2.) I disagree that all organs are fully functional within 48 hpf. For example, the larval swim bladder is not inflated until 3-4 dpf. I would change the wording to say that most organ systems are functional at 48 dpf (lines 48-49).
3.) In section IV.3., the study by Crilly et al (which was cited in the introduction) should be referenced as their high-throughput screen of small molecules administered by bath immersion identified drugs which prevented brain cell death in a zebrafish larval model of ICH, and highlights the utility of this approach.
4.) There are more recent papers published on adult zebrafish oral administration which may be appropriate to be referenced in section III.1.3 e.g. Aleksander J. Ochocki, Justin W. Kenney; A gelatin-based feed for precise and non-invasive drug delivery to adult zebrafish. J Exp Biol 15 January 2023; 226 (2): jeb245186. doi: https://doi.org/10.1242/jeb.245186; Lu, Y., & Patton, E. E. (2022). Long-term non-invasive drug treatments in adult zebrafish that lead to melanoma drug resistance. Disease models & mechanisms, 15(5), dmm049401. https://doi.org/10.1242/dmm.049401
5.) Figure 1 is a good visual summary of the drug administration methods, but the labels ‘Hydrogels’, ‘Nanoparticles’ and ‘Genetic Manipulation’ should be rotated so they are easier to read (lines 154-156).
6.) Additionally, there are a few parts of the text that should be checked:
Several words have been capitalised when they shouldn’t e.g zebrafish (many times including line 160), lithium (line 506) and iPSCs (lines 690 and 696).
Gene names should be italicised (lines 534 & 535).
Correct Liu et al (line 662).
The quality of the English is good.
Reviewer 2 Report
The review recaps the knowledge on the various drug delivery techniques and therapeutics administration to zebrafish, as a model for neurodegenerative diseases. Due to the more and more difficulties in using rodent models for ethical and founding reasons, joined to the easier manipulation and the high number of subject shortly available, zebrafish is a growing approach in neurodegenerative diseases research. Despite the Authors nicely and critically resumed the information on the various drug delivery techniques in the model, the review looks to me too little for a journal with an impact factor of 5.6. Some suggestion to increase the content of this manuscript accordingly with the quality (and quantity) of IJMS is given.
In general the review is well written; in section I, the pros are well listed and referenced; and section III, “drug administration” is nice and exhaustive.
Despite I understand that section III is the goal of the review, in agreement with what the authors stated, the amount of information actually presented by the review is little for a journal like IJMS. To fill the gap, additional information may be added to section II and IV.
In detail:
Section II, is little developed, and may be improved by adding content on:
- Even if the topic has already been discussed in recent reviews, a description or a table resuming the main features (reached targets, and unreached targets) of the major neurodegenerative diseases in humans, vs rodents, vs zebrafish has to be added.
- Chemicals (for inducing a disease) may induce very different response based on the stage of brain development at the time of challenging. A tentative parallel among human, rodent (as models), and zebrafish brain development may be helpful and relevant to section IV.
- Chance to collect enough (or limitation in) amount of sample (protein, mRNA, genome) for biomolecular analysis, especially if a specific region of the brain has to be studied, considering that large part of neurodegenerative diseases affect specific structures, and cells.
- Comparison of the pathologic mechanism among zebrafish and rodent models (e.g.: inflammation, oxidative-stress, neurotransmitter alterations, developmental alterations, hypotrophy, etc.), agreement or disagreement (still text or table). This is central to understand the real advantages of performing drug screening in zebrafish, thus is a needed point to discuss before moving to the methods for delivering drugs to the animals.
- Drug screening of candidate compounds for a diseases in Zebrafish, vs rodents. There is data on the inter-models reproducibility?
Section IV: add some disease to AD, PD, TBI & stroke, ASD. As example, zebrafish has been used to model Huntington disease, amyotrophic lateral sclerosis (as mentioned in the introduction), multiple sclerosis, pontocerebellar hypoplasia, heavy metals toxicity, spinocerebellar ataxia, pediatric-onset neurological disorder.
Round 2
Reviewer 2 Report
I appreciated the changes done by the authors. The review has improved substantially. I suggest adding still a table resuming the major pro and cons of Zebrafish vs rodent models in mimicking the major neurodegenerative /neurologic conditions, as previously suggested. Even a short table referring to few of the nice reviews that have been published in the next years. This will definitively make the review complete and very nice

Author Response
We appreciate the reviewer’s comments that helped us substantially revise the manuscript. However, we disagree with the necessity to add extensive information about the different neurological models existing in mice vs zebrafish. This is not the scope of our review that aimed to discuss drug delivery techniques and systems. The comparison between all the models requires precise analysis of the outcomes of each model and would make on its own an excellent topic for a future review pertaining models of neurodegenerative diseases across species. Nevertheless, we added some references of excellent recent reviews about this topic in the manuscript (Lines 127-128).